# The Spike Protein of SARS-CoV-2 Is Adapting Because of Selective Pressures

**DOI:** 10.3390/vaccines10060864

**Published:** 2022-05-28

**Authors:** Georgina I. López-Cortés, Miryam Palacios-Pérez, Hannya F. Veledíaz, Margarita Hernández-Aguilar, Gerardo R. López-Hernández, Gabriel S. Zamudio, Marco V. José

**Affiliations:** 1Theoretical Biology Group, Instituto de Investigaciones Biomédicas, Universidad Nacional Autónoma de México, Mexico City 04510, Mexico; mir.pape@iibiomedicas.unam.mx (M.P.-P.); hanny.v.fuentes@gmail.com (H.F.V.); mhernandezaguilartheorbiol@gmail.com (M.H.-A.); dolphin@ciencias.unam.mx (G.R.L.-H.); gazaso92@gmail.com (G.S.Z.); 2Network of Researchers on the Chemical Evolution of Life, NoRCEL, Leeds LS7 3RB, UK

**Keywords:** SARS-CoV-2, variants of concern, S protein, receptor binding domain, neutral substitution test, structural analysis, hydrophobic and electrostatic potentials

## Abstract

The global scale of the COVID-19 pandemic has demonstrated the evolution of SARS-CoV-2 and the clues of adaptation. After two years and two months since the declaration of the pandemic, several variants have emerged and become fixed in the human population thanks to extrinsic selective pressures but also to the inherent mutational capacity of the virus. Here, we applied a neutral substitution evolution test to the spike (S) protein of Omicron’s protein and compared it to the others’ variant of concern (VOC) neutral evolution. We carried out comparisons among the interactions between the S proteins from the VOCs (Alpha, Beta, Gamma, Delta and Omicron) and the receptor ACE2. The shared amino acids among all the ACE2 binding S proteins remain constant, indicating that these amino acids are essential for the accurate binding to the receptor. The complexes of the RBD for every variant with the receptor were used to identify the amino acids involved in the protein—protein interaction (PPI). The RBD of Omicron establishes 82 contacts, compared to the 74 of the Wuhan original viral protein. Hence, the mean number of contacts per residue is higher, making the contact thermodynamically more stable. The RBDs of the VOCs are similar in sequence and structure; however, Omicron’s RBD presents the largest deviation from the structure by 1.11 Å RMSD, caused by a set of mutations near the glycosylation N343. The chemical properties and structure near the glycosylation N343 of the Omicron S protein are different from the original protein, which provoke reduced recognition by the neutralizing antibodies. Our results hint that selective pressures are induced by mass vaccination throughout the world and by the persistence of recurrent infections in immunosuppressed individuals, who did not eliminate the infection and ended up facilitating the selection of viruses whose characteristics are different from the previous VOCs, less pathogenic but with higher transmissibility.

## 1. Introduction

In the past few months (2021–2022), mankind has suffered new waves of highly transmissible variants of SARS-CoV-2. The latest variant of concern (VOC) was first isolated in Botswana and South Africa on 11 November 2021, and it was named Omicron (B.1.1.529). It took only 17 days after the first isolation (26 November 2021) for the World Health Organization to classify it as a VOC [1].

Its high transmissibility resulted in its fast worldwide spread and by 22 January 2022, 150 countries confirmed infections with the new variant [2]. It also rapidly became the dominant variant in South Africa, the United Kingdom, Denmark, India, and the United States [2,3,4,5]. Omicron is subdivided into three sub-lineages BA.1, BA.2, BA.3 that carry 51 mutations all over the genome; in the spike (S) protein, they only present between 31 and 37 mutations, inscluding insertions and deletions [6]. Differences in the amino acid sequence are the most accepted signal that may herald changes in transmissibility, reinfection, susceptibility, and immune evasion. Specifically, the sub-lineage BA.2 was found to be the most transmissible, whereas BA.3 was the least [3,4,5].

Owing to the large number of mutations in the spike protein and elsewhere on the virus [7], there is concern that Omicron will exhibit substantial escape from vaccine-elicited immunity [8,9,10].

Most vaccines against SARS-CoV-2 have been developed using the S protein as the antigen to induce the immune memory [11,12,13,14], which is actually comparable to the immune memory generated by natural infections [13,14]. Regardless of the route of immunization, the production of neutralizing antibodies has an important role for effectively clearing the virus before it infects the cells. This is because the main target of neutralizing antibodies is the receptor binding domain (RBD) of the S protein [9]. At the same time, the RBD is under high selective pressures to adapt to the receptor. Thus, both the neutralizing antibodies and the higher affinity to the receptor are important competing selective pressures of viral evolution.

To unravel the evolutionary mechanisms of the S protein of SARS-CoV-2, we carried out an evolutionary neutrality test [15] for the S protein of the Omicron variant and compared it to the neutrality test calculated for a previous set of S proteins of SARS2-S VOC sequences [16]. The mutations of each VOC at the RBD until the furin cleavage site were also examined. Then, the complexes of the RBD of every variant with the receptor ACE2 were used to identify the amino acids involved in the protein—protein interaction (PPI). The properties of the contact amino acids were analyzed in terms of the electric potential and hydrophobic potential of the interface of the RBDs of each VOC. Additionally, we calculated the deviations in the structures of the RBD of each VOC compared to the Wuhan original structure. Finally, we looked for changes in the amino acids near the glycosylation sites from the reference structure and of Omicron’s RBD. Consistent with our previous work, the amino acids shared among all the ACE2 binding S proteins remain constant, indicating that they are essential for the accurate binding to the receptor. The remaining amino acids are responsible for changes in the binding affinity to the receptor and immune evasion. The results are discussed in terms of its implications for mass vaccination and immune response.

## 2. Data Sources

The nucleotide and amino acid sequences of 100 Omicron SARS2-S reported in the GenBank SARS-CoV-2 Data Hub (https://www.ncbi.nlm.nih.gov/labs/virus/vssi/#/sars-cov-2, accessed on 3 April 2022) were obtained. The structures of the RBD of SARS2-S of each variant in complex with ACE2 were downloaded from the Protein Data Bank (PDB, https://www.rcsb.org/, accessed on 3 April 2022), with the following codes: 6M0J denotes the reference RBD + ACE2 isolated from Wuhan; 7EKF is the Alpha variant RBD + ACE2 complex; 7VX4 is the Beta variant RBD + ACE2 complex; 7EKC is the Gamma variant RBD + ACE2 complex, 7WBQ is the Delta variant RBD + ACE2 complex; and 7WBP is the Omicron variant RBD + ACE2 complex.

## 3. Materials and Methods

### 3.1. Neutral Evolution Test

As described elsewhere [15], a neutral evolution test was performed for every group of 100 sequences of each variant and for 100 sequences reported until June 2020. We built another group, taking sequences from all six groups. Our amino acid neutrality test considers that all positions have the same probability to mutate, with the sole constraint that they obey the degeneracy of the SGC. Briefly, for every set of sequences, they were pairwise aligned using MUSCLE [17] using MEGA 11 software [17,18] using the default parameters. The alignment was used to generate a table of mutations that calculates the number of changes in codons. This table is transformed into an amino acid substitution matrix, which is then normalized to generate a probability transition matrix, from which a stationary probability distribution is obtained. The resulting calculation reveals the probability of occurrence of each amino acid in the protein if each amino acid has a uniform distribution of mutation. This model gives the probability of occurrence of each amino acid for a given protein and indicates the type of selective pressure exerted in each type of amino acid. If the probability of occurrence for an amino acid has a greater value than its corresponding value at the neutral control (a hypothetical protein subjected only to random changes), it will be interpreted as positive selection, and lower values will be considered as negative selection. The values that lie at or near to the neutral control will be interpreted as consistent with neutral or nearly neutral mutations. To assess the statistical robustness of the sample of sequences, a jackknife procedure was applied. A confidence interval of 95% was computed around the stationary distribution derived from the set of sequences.

### 3.2. Structural Analysis

The protein structures were downloaded from the PDB and they were cleaned and the reported glycosylations were considered. The structural analyses were visualized and analyzed with Chimera and Chimera X [19,20]. The complexes of the RBD for every variant with the receptor were used to identify the amino acids involved in the PPI. The distances, chemical interaction, the hydrophobicity and electrostatic potentials were computed for the RBDs’ surface residues at the interface.

## 4. Results

We carried out a neutrality test for the S protein of the Omicron variant (Figure 1) and compared it to the neutrality test calculated for a set of S proteins of SARS-CoV-2 sequences reported until June 2021 (data gathered from [16]). Then, we described the type of selective pressure, positive or negative, that each mutation had in the RBD for each VOC (Table 1). In fact, all the mutations in the RBD of the VOCs were under selective pressures except for one position, meaning that the probabilities of occurrence of the amino acids associated to those mutations deviated from the expected probability by random neutral mutations. Importantly, Figure 1 shows that most of the amino acids of Omicron S protein have the same type of selection as the previous SARS2-S proteins, but the probabilities deviate less from the neutral control. Only Cys, Glu, and Gly of Omicron’s S have a different type of selective pressure than the previous SARS2-S variants.

We analyzed the physicochemical properties of the residues of the RBD of the SARS-CoV-2 variants to explain the differences in binding affinity to the receptor. To this end, we compared the previously reported properties of the original Wuhan isolated variant RBD in complex with the receptor ACE2 with those of the VOCs’ RBDs. As reported, SARS2-S establishes contacts with the receptor by means of 17 residues, from which 7 residues are conserved among the hACE2 binding S proteins and the 10 unique residues responsible for increasing the binding affinity [16]. Here, we identified that those conserved amino acids present in all the ACE2 binding S proteins remain constant for all VOCs, except for the position T505. Nevertheless, not all of them establish contact with the receptor. Indeed, we found that the residues involved in the binding may be altered. The residues F456, A475, F486, N487, Y489, T500 and G502 are used for all the VOCs for the binding, while positions 493, 498, 501 and 505 remain necessary for the binding even if they have mutated in at least one VOC (Table 2). The contacts between residues K417, G446 and Y449 and the receptor are only formed in a few VOCs, and their contribution to the binding is non-essential.

The number of contacts or even the number of residues interacting with the receptor’s residues does not seem to be determinant for the binding affinity, but the kind of interactions does. The Beta variant has a unique interaction because the mutations do not favor the formation of hydrogen bonds (there are only two) but do favor the hydrophobic interactions (Table 3). Interestingly, the Omicron and Delta variant form up to 13 and 12 hydrogen bonds, and 12 and 10 hydrophobic contacts, respectively.

Of the 26–32 mutations in Omicron’s S protein, 16 are in the RBD. Of those, five substitutions use polar and positively charged amino acids (Table 1), which change the properties of the interface with the receptor. When comparing the electric potential of the interface, Omicron clearly has positive spots, whereas the original structure and the other VOCs present polar but neutrally charged amino acids at the interface (Figure 2a). Omicron also displays a lower hydrophobic potential; nevertheless, there are conserved spots that do not change (Figure 2b).

The structure of the RBD of the VOCs is still quite similar despite all the mutations. As expected, Omicron’s RBD has the highest deviation compared with the Wuhan’s RBD, 1.11 Å RMSD. The other VOCs overlap much more with the Wuhan structure, showing 0.55 Å, 0.54 Å, 0.74 Å, and 0.55 Å RMSD for Delta, Gamma, Beta and Alpha, respectively (Table 4). The RMSD reflects that all the mutations in the Omicron variant change not only the sequence but also the structure, which compromise antibody recognition. Figure 3 illustrates the overlapping of all the VOCs’ RBDs and the Wuhan’s RBD, and it can be noticed that the Omicron variant is the most discrepant.

Only Omicron’s RBD presents mutations near a glycosylation site, specifically near N343. In fact, these set of mutations changed the entire glycosylation surroundings. These mutations caused the NAG that comes in contact with four residues from the same RBD chain, instead of only two residues as it occurs in other variants (i.e., F338, D339, F342 and N370, and with G339 and F338, respectively) (Figure 4). The RBD of the previous variants presented an interaction between S373 and S375, avoiding the interaction of N370 with the carbohydrate. In the Omicron RBD, the mutation P373S causes the interaction between Serines to break and also that they approaching to N370. Additionally, interactions within the trimer could be disturbed, and could alter epitopes that would be hardly detected by neutralizing antibodies.

Surprisingly, the Omicron variant has around 30 amino acids substitutions and 6 deletions in the S protein. Of all the mutations, 16 are in the RBD and correspond to amino acids under selective pressure, which means that the protein underwent Darwinian adaptation. A notorious example is that the RBDs’ surface at the interface changed the electrostatic properties, from neutral to positive. The receptor’s surface presents negatively charged spots (Figure 5), such that both surfaces attract each other, increasing the binding affinity.

## 5. Discussion

All the mutations in the RBD of VOCs have been fixed by natural selection because the used amino acids do not rely upon the expected probability of stochastic mutations. Table 1 shows that all the substitutions use amino acids under either positive or negative selective pressure. This means that the new amino acids have been selected because they contribute to the adaptation to the receptor and/or for evading the host’s immune response.

All those mutations are still susceptible to further mutations, including neutral mutations, but selective pressure could fix new ones. Meanwhile, the VOCs’ sequences persist with possible polymorphisms. The mutational profile of Omicron’s spike exhibits Darwinian evolution but tends to be closer to the neutral control and is more similar to the spike of *Betacoronavirus* than the previous variants Alpha, Beta, Gamma, and Delta. In contrast, the evolutionary mechanism of the RBD clearly follows Darwinian adaptation.

The probabilities computed from the neutrality test of the Omicron S protein deviate less from the neutrality test applied to the S proteins gathered before June 2021. This means that the S protein, in this time frame of an emerging virus, is still being molded. In fact, Omicron S protein probabilities resemble even more the probabilities obtained for the S proteins of the *Betacoronavirus* genera [16]. To note, those probabilities reflect the selective pressures of the coronavirus S protein over years of evolution. This phenomenon may be explained by the radiation of variants when an emerging virus adapts to a new host. SARSCoV-2 exhibited, initially, large deviations from neutral mutations until the probabilities of occurrence became similar to those of homologue proteins living in similar conditions, which are also understood as being exposed to similar selective pressures.

The mutations of Omicron’s RBD contribute to the increase in the binding affinity for the receptor. Firstly, the RBD establishes 82 contacts, in comparison to the 74 found in the Wuhan original viral protein, but uses 16 contacts residues instead of 17. Hence, the mean number of contacts per residue is higher, making the contact thermodynamically more stable. Other mutations are found on the RBD surface on one side, and these interact with the carbohydrates at the glycosylation site N343, changing both the structural conformation of the peptide and the chemical properties. In addition, the mutations at the RBDs’ interface change the electrostatic properties, from neutral to positive. The receptor’s surface presents negatively charged spots (Figure 5), such that both surfaces attract each other, increasing the binding affinity.

The number of contacts for each amino acid involved in the binding is not the same but most of the sequences in the RBD are conserved. Indeed, there are amino acids among the ACE2 binding S proteins [16] that remain the same. This proves that these amino acids were already fixed by natural selection. Importantly, some specific mutations alter the balance of binding affinity, for example, substitution K417N in Beta, Gamma, and Omicron variants does not favor binding because N417 does not form the hydrogen bond that the ancestral sequence does, resulting in the loss of contact with the receptor. On the contrary, N501Y substitution increases the binding to the receptor [21,22,23] by an increase in the number of contacts. The substitution D614G in all VOCs and in most lineages is explained by common selective pressures that drove its fixation. This mutation enhances the open conformation of the S protein, increasing the probability of interaction with the receptor [24]. The epidemiological consequence was that D614G increased the transmissibility of the virus [25,26,27].

The immune system is one of the major forces that compels the virus to evolve. In order to counteract this force, the S protein is protected by a glycoside shield [28,29]. It is calculated that SARS2-S is covered by around 40% of its surface by carbohydrates [29,30]. Nevertheless, the VOCs of SARS-CoV-2 constantly present mutations near the glycosylation sites, which alter the surface of the glycoprotein and, therefore, the epitopes recognized by antibodies [29]. The chemical properties and structure near the glycosylation N343 of the Omicron S protein are different from the original protein. A series of mutations near N343 cause the largest deviation in the structure of the RBD of the protein in comparison to the rest of the RBDs (Figure 3). Consequently, the recognition of neutralizing antibodies is reduced. Additionally, the structural analysis of the trimer of Omicron’s S protein revealed that mutations in the positions S371, S373 and S375 enhance the accurate packing and stabilization of the RBD–RBD interaction [31]. These residues were identified to stabilize the RBD–RBD interaction in the RBD-down conformation with R403, Q493 and Y505 from the next protomer, while the third protomer remains in the open conformation [32].

Various mutations in the VOCs’ S protein escape the recognition of antibodies produced by the host. This strategy is well documented for mutations in hot spots where specific antibodies bind [10]. Furthermore, if a mutation alters a hot spot in the RBD, the neutralizing antibodies will not be able to recognize the protein; therefore, the virus would infect the cells as if there was not previous immunization/vaccination. The mutations E484K and the E484A, present in the Beta, Gamma, and Omicron variants, occur in amino acids with different selection but both mutations lead to neutralizing antibody resistance [33]. A study using amino acid interaction networks suggested that mutations in the Delta and Beta variants disturb the recognition of therapeutic non-neutralizing antibodies, whereas Omicron’s mutations reduce the recognition of neutralizing (i.e., N440K, G446S, G496S and Q498R) and nearly neutralizing (G339D, S371L, S373P and S375F) antibodies [34]. This study predicted that therapeutic antibodies would fail in recognizing the S protein. The best strategy is not to use monoclonal antibodies that bind to specific epitopes that are under selective pressures but to use polyclonal antibodies and/or to elicit the host’s B cell response that could recognize different epitopes along the virus, especially those involving amino acids that are crucial for the binding to the receptor or conserved regions.

The mass vaccination campaigns have protected entire populations from further infections, and the race of further viral mutations have been diminished. The susceptibility to developing severe disease is higher in unvaccinated people [35]. This is more of a concern in immunosuppressed/immunocompromised individuals, who are vulnerable to any infection including SARS-CoV-2, but they are considerably protected with the vaccines [36]. Moreover, various studies have shown that a booster vaccine against SARS-CoV-2 significantly reduces the susceptibility and increases the immune protection toward any variant, including Omicron [3,37,38,39]. Although some vaccines were shown to reduce the effectiveness against Omicron, the reality is that they are still protecting by inducing a significant humoral and cellular immune response. T cells recognize both conserved and new epitopes, making the response robust and buffering the mutations that evade neutralizing antibody recognition [11,40].

Mutations in the Omicron S protein have conferred advantages in the binding and immune escape but the physiological consequences seem to benefit the host too. Unlike the other variants, the Omicron variant replicates more efficiently in the bronchial tubes rather than in the lung tissue [41]. Other VOCs infect equally the bronchi and the lung cells, just as the original virus does. It is not completely clear whether the preference of the infection for the bronchi is due to characteristics of the host or the infective capacity of the virus. For example, on the host side, the level of expression of ACE2 and TMPRSS2 is higher in the bronchi. On the other side, the Omicron variant fails to infect cells expressing low levels of ACE2 and TMPRSS2 [42]. Remarkably, the higher affinity of Omicron’s S protein to the receptor may also be key for stopping or delaying the viral particles in upper tissues with greater expression of this receptor. To complete this idea, there is also a need for studies revealing the physiological response in the upper respiratory tract and the immunological response in mucosa. Nevertheless, the selected mutations of Omicron’s proteins have caused the preference of bronchi over the lung; therefore, this leads to a lower number of inflamed tissues, reduction in organ misfunction, as well as other complications and mortality.

The Omicron S protein is cleaved slightly slower than the other VOC, mainly because (i) the mutations near the furin site reduced the proteolytic activity of the enzyme and (ii) the N679K mutation contributed to the less disordered structure by establishing contacts with the C terminal domain [42]. The Omicron variant is less dependent on TMPRSS2 for internalization [41]. Further analysis of the viral entrance would reveal, if any, different internalization mechanisms.

## 6. Conclusions

The strong selective pressure exerted on the virus has selected mutations that have increased its transmissibility [24]. Shared mutations among different variants arose by convergent evolution caused by common selective pressures regarding viral fitness by increasing the pathogenicity, PPI, and the evasion of the immune system [43]. Nevertheless, the huge number of mutations in the Omicron variant is in the spotlight. The fact that the mutations have allowed the S protein to have a higher affinity for the receptor can be understood in the light of adaptation by natural selection. The selected characteristics could have been the result of selection pressures induced by mass vaccination throughout the world and by the persistence of recurrent infections in immunosuppressed/immunocompromised individuals, who did not eliminate the infection and ended up facilitating the selection of viruses with characteristics different from the previous VOCs, but with the same or greater affinity.

To explain how the Omicron variant emerged with such a large amount of mutations, different hypotheses have been propounded [2]. The ancestral lineage may have been evolving (i) in isolated populations, (ii) in immune-compromised individuals, (iii) in infected animals and returned to humans as a zoonotic event, or (iv) by the recombination of different lineages. Although for now the origin is not known, the Omicron ancestral virus evolved independently, forming a different monophyletic clade [44,45,46]. Some mutations were selected in order to better adapt to the host; however, there must have been an event that enhanced the mutation rate notwithstanding the most related variant, Alpha, which had less mutations and a lower mutational rate [44]. The most probsable situation was that the Omicron ancestral virus emerged a long time ago and it has been circulating and evolving without being identified but under stringent selective pressures. The neutrality test of Omicron’s S protein announces the probable start of the end of the pandemic, indicating adaptation to the host. The more close the probabilities of occurrence get to the ones of the S proteins of the *Betacoronavirus* [16], the more the protein approaches to the long lasting behavioral evolution of viral proteins, which means a period of adaptation to the host.

## Figures and Tables

**Figure 1 vaccines-10-00864-f001:**
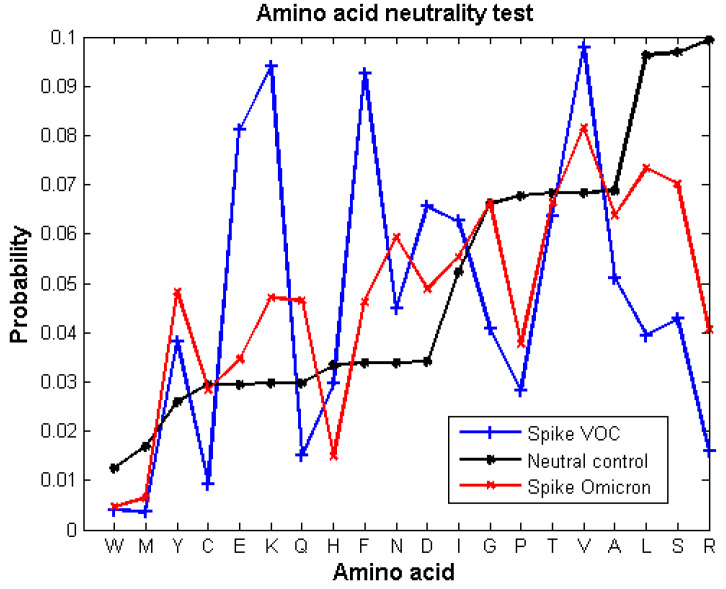
Neutrality substitution test from the S protein of Omicron variant (red line) compared to the previously reported neutrality test of the SARS2-S VOC sequences reported before June 2021 (blue line) (data obtained from our previous work [16]). The negative control (black line) shows the expected probability of amino acids occurring by random mutations. The probability of occurrence of each amino acid of the proteins was calculated with a confident interval of 95%. Amino acids showing a probability of occurrence different from the expected control are under selective pressures, either positive or negative.

**Figure 2 vaccines-10-00864-f002:**
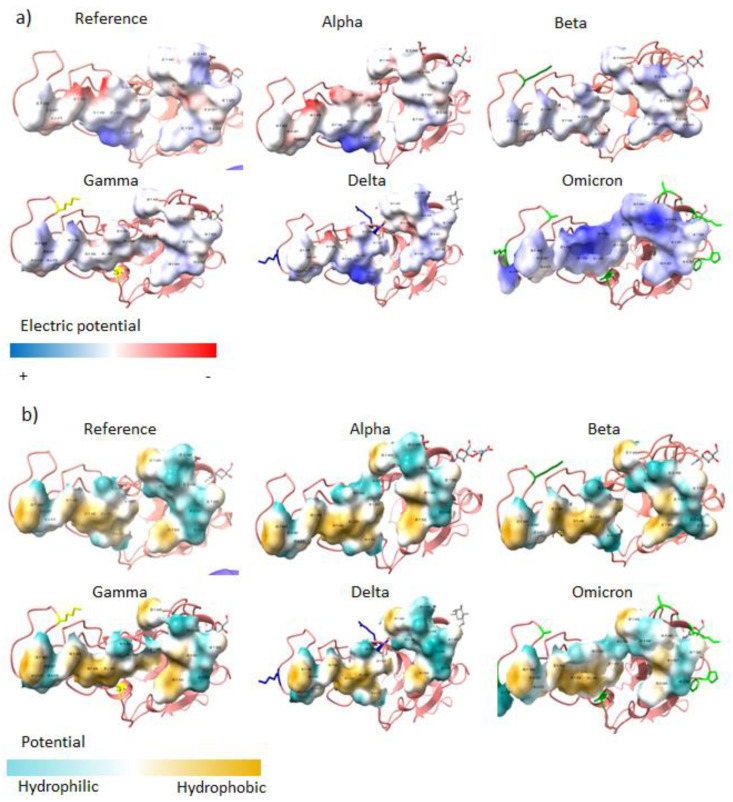
Interface of the RBD of each VOC. (**a**) Electric potential and (**b**) hydrophobic potential of the interface of the RBDs of each VOC.

**Figure 3 vaccines-10-00864-f003:**
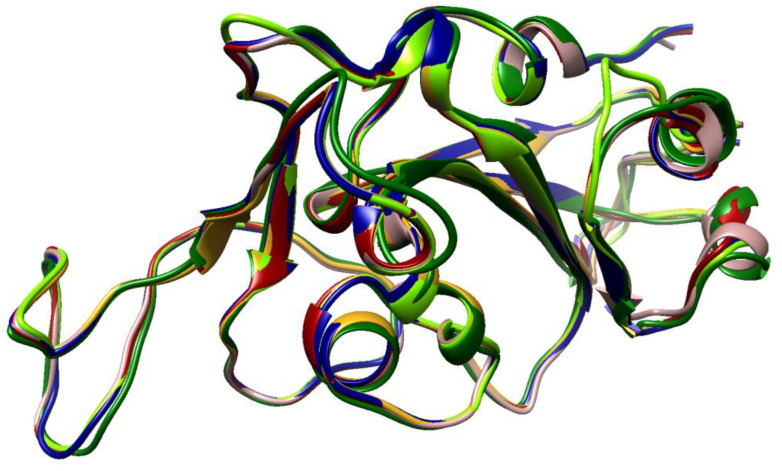
VOC RBD overlapping. Typical view of the orientation of the RBD of each VOC when it is coupled with the receptor. Rosy, brown—reference original structure, dark red—Alpha, dark green—Beta, golden rod—Gamma, navy blue—Delta, and chartreuse—Omicron.

**Figure 4 vaccines-10-00864-f004:**
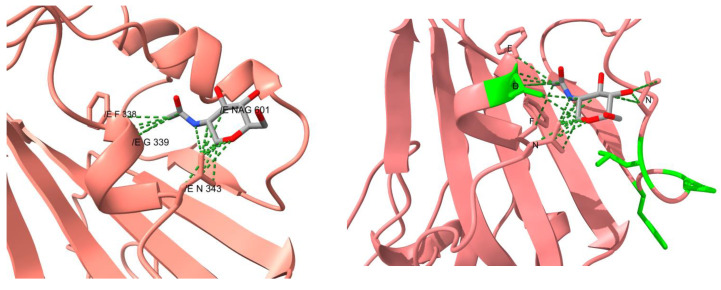
Comparison of the contacts at the glycosylation site N343 from the reference structure and of Omicron’s RBD. The carbohydrate in the reference structure is colored in light blue. The residues highlighted in green lime in the RBD of Omicron are the mutated residues.

**Figure 5 vaccines-10-00864-f005:**
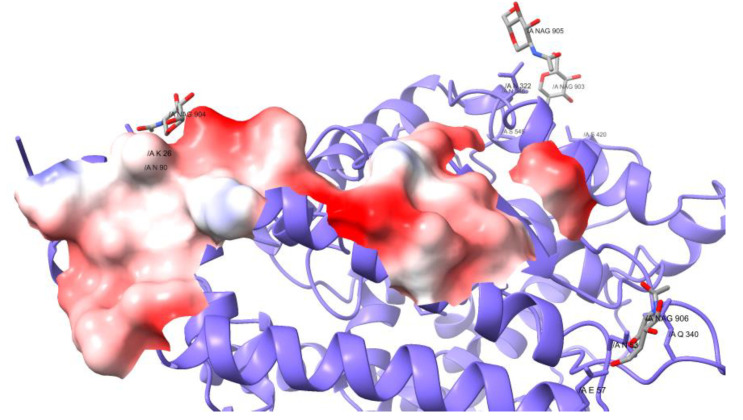
Electrostatic potential of the receptor ACE2 in the interacting surface with SARS2-S.

**Table 1 vaccines-10-00864-t001:** Mutations of each VOC at the RBD until the furin cleavage site. Mutations are shadowed by the chemical characteristic of the residue: yellow—non-polar, green—polar uncharged, blue—polar positively charged and red—polar negatively charged. The symbol next to each residue refers to the type of selective pressure exerted over that amino acid; thus, + is positive selective pressure, - is negative selective pressure, and / is for neutral selection.

Reference		Alpha		Beta		Gamma		Delta		Omicron	
339	G									D	+
371	S									L	-
373	S									P	-
375	S									F	+
417	K			N	+	T	+			N	+
440	N									K	+
446	G									S	-
452	L							R	-		
477	S									N	+
478	T							K	+	K	+
484	E			K	+	K	+			A	/
490	F										
493	Q									R	-
496	G									S	-
498	Q									R	-
501	N	Y	+	Y	+	Y	+			Y	+
505	T	Y	+			Y	+	Y	+	H	-
547	T									K	+
570	A	D	+								
614	D	G	-	G	-	G	-	G	-	G	-
655	H					Y	+			Y	+
677	Q										
679	N									K	+
681	P	H	-					R	-	H	-

**Table 2 vaccines-10-00864-t002:** Contact residues of the Wuhan original sequence of SARS-CoV-2 and the VOCs with the receptor ACE2. The list of the contact residues for every RBD is listed. The number at the right of each residue is the number of contacts of that specific residue. The underlined position refers to conserved positions from ACE2 binding to S proteins. The shadowed positions are contact residues that have not mutated in VOCs and the red residues are the mutated contact residue.

Reference			Alpha		Beta		Gamma		Delta		Omicron	
417	K	4	K	1					K	3		
446	G	1										
449	Y	6							Y	5	Y	2
453	Y	1			Y	3	Y	1	Y	3	Y	3
455	L	2			L	3	L	4	L	2		
456	F	5	F	4	F	8	F	5	F	2	F	3
473	Y		Y	1	Y	1						
475	A	2	A	2	A	1	A	1	A	1	A	2
476	G				G	1					G	1
486	F	9	F	5	F	8	F	6	F	5	F	7
487	N	3	N	7	N	5	N	2	N	6	N	4
489	Y	4	Y	4	Y	5	Y	5	Y	4	Y	6
493	Q	2	Q	13	Q	6	Q	2	Q	3	R	7
494	S		S	2					S	2	S	2
496	G	1							G	2	S	2
498	Q	7	Q	3	Q	3	Q	4	Q	2	R	4
500	T	9	T	8	T	8	T	10	T	7	T	6
501	N	1	Y	12	N	5	Y	14	N	9	Y	14
502	G	5	G	5	G	1	G	2	G	4	G	6
503	V				V	1						
505	Y	12	Y	10	T	13	Y	16	Y	20	H	13

**Table 3 vaccines-10-00864-t003:** Properties of the interaction between RBDs of the VOCs and the receptor.

	SARS-CoV-2	Alpha	Beta	Gamma	Delta	Omicron
Hydrogen bonds	10	8	2	6	12	13
Hydrophobic contacts	18	11	21	16	10	12
Contacts	74	77	72	72	80	82
Mean distance (Å)	3.563	3.572	3.414	3.569	3.492	3.557
Number of residues	17	14	16	13	17	16

**Table 4 vaccines-10-00864-t004:** Deviations in the structures of the RBD of each VOC compared to the Wuhan original structure.

	RMSD (Ǻ)	TM Score
Alpha	0.55	0.99
Beta	0.74	0.98
Gamma	0.54	0.99
Delta	0.55	0.99
Omicron	1.11	0.97

## Data Availability

Not applicable.

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
