# Peer review of "The Spike Protein of SARS-CoV-2 Is Adapting Because of Selective Pressures"

_vaccines, 2022, doi:10.3390/vaccines10060864_

Round 1

Reviewer 1 Report

The author has reported mass vaccination throughout the world and the persistence of recurrent infections in immunosuppressed individuals, who did not eliminate the infection and ended up facilitating the selection of viruses whose characteristics are different from the previous VOCs, less pathogenic but with higher transmissibility. This Manuscript is well written and needs minor revision.

Comments:

  1. Every section of the manuscript must be written scientifically according to the published literature with appropriate references.
  2. The logical flow of this manuscript is not perfect. The authors have written several matters haphazardly. The work appears as groundwork.
  3. Spacing, punctuation marks, grammar, and spelling errors should be reviewed wholly.
  4. The flow of the introduction is not complete and unspecific. My recommendation is to construct the sentences more lucid and legible for more productive comprehension.
  5. The first paragraph of the introduction section contains no new information. Need to change.
  6. The introduction section seems missing important information. This section needs profound modification with different medical hypotheses recommended different treatment approaches and gave convincing arguments for new discoveries urgently needed for treatment options COVID-19. Authors can also write some about the recent mutation.
  7. RMSD of Alpha, Beta Gamma, and Delta are near about the same but Omcron has more deviated. What is the meaning of that in table 4.
  8. Page no. 7, the author has mentioned “This study predicted that therapeutic antibodies would fail in recognizing the S protein. Which strategy is best ?.

Author Response

May 13th, 2022

Mr. Aiden Yu
Editor vaccines

E-Mail: aiden.yu@mdpi.com

Manuscript ID: vaccines-1688687 

Title: The spike protein of SARS-CoV-2 is adapting because of selective pressures

Authors: Georgina I. López-Cortés, Miryam Palacios-Pérez, Hannya F. Veledíaz, Margarita Hernández-Aguilar, Gerardo R. López-Hernández, Gabriel S. Zamudio, Marco V. José,

Dear Dr. Aiden Yu:

Thank you for your email dated May 10th, 2022, and for the reviewer’s comments.

We start by thanking the reviewers for their careful reading of our paper and for helping us to improve its quality and presentation.

Please find enclosed with this letter the revised version of the manuscript.

We have carefully read the reviewer’s comments of our previous version of the manuscript, and we have rewritten the paper with their suggestions in mind. The reviewers raised a series of good questions. We have considered their comments modifying the paper accordingly. The modifications in the manuscript are highlighted in red color, and the answers in this reply letter are also marked in red.

We give below the specifics of our response to the reviewers and answer their criticisms in the same order as their reports.

REVIEWER 1

The author has reported mass vaccination throughout the world and the persistence of recurrent infections in immunosuppressed individuals, who did not eliminate the infection and ended up facilitating the selection of viruses whose characteristics are different from the previous VOCs, less pathogenic but with higher transmissibility. This Manuscript is well written and needs minor revision.

Comments:

  1. Every section of the manuscript must be written scientifically according to the published literature with appropriate references.
  2. The logical flow of this manuscript is not perfect. The authors have written several matters haphazardly. The work appears as groundwork.
  3. Spacing, punctuation marks, grammar, and spelling errors should be reviewed wholly.
  4. The flow of the introduction is not complete and unspecific. My recommendation is to construct the sentences more lucid and legible for more productive comprehension.
  5. The first paragraph of the introduction section contains no new information. Need to change.
  6. The introduction section seems missing important information. This section needs profound modification with different medical hypotheses recommended different treatment approaches and gave convincing arguments for new discoveries urgently needed for treatment options COVID-19. Authors can also write some about the recent mutation.
  7. RMSD of Alpha, Beta Gamma, and Delta are near about the same but Omcron has more deviated. What is the meaning of that in table 4.
  8. Page no. 7, the author has mentioned “This study predicted that therapeutic antibodies would fail in recognizing the S protein. Which strategy is best?

We thank the reviewer for her/his comments. We agree with the reviewer that Introduction lacked structure and relevant information. Therefore, we added some paragraphs to improve the flow of the Introduction.

Response to comments 1-6:

“Most vaccines against SARS-CoV-2 have being developed using the S protein as the antigen to induce the immune memory [10-11] which is actually comparable to the immune memory generated by natural infections [12-13]. Regardless the route of immunization the production of neutralizing antibodies has an important role for effectively clearing the virus before it infects the cells. This is because the main target of neutralizing antibodies is the Receptor Binding Domain (RBD) of the S protein [8]. At the same time the RBD is under high selective pressures to adapt to the receptor. Thus, both the neutralizing antibodies and the higher affinity to the receptor are important competing selective pressures of viral evolution.”

“To unravel the evolutionary mechanisms of the S protein of SARS-CoV-2, we carried out an evolutionary neutrality test [15] to the S protein of the Omicron variant and compared it to the neutrality test calculated for a previous set of S proteins of SARS2-S VOC sequences [16]. Mutations of each VOC at the RBD until the furin cleavage site were also examined. Then the complexes of the RBD of every variant with the receptor ACE2, were used to identify the amino acids involved in the protein- protein interaction (PPI). The properties of the contact amino acids were analyzed in terms of the electric potential and hydrophobic potential of the interface of the RBDs of each VOC. Additionally we calculated the deviations of the structures of the RBD of each VOC compared to the Wuhan original structure. Finally, we looked for changes of the amino acids near the glycosylation sites from the reference structure and of Omicron’s RBD. Consistent with our previous work, the amino acids shared among all the ACE2 binding S proteins remain constant, indicating that they are essential for the accurate binding to the receptor. The remaining amino acids are responsible for changes in the binding affinity to the receptor and or immune evasion. The results are discussed in terms of its implications of mass vaccination and immune response.”

Background work paragraphs have been omitted:

“Just as the Omicron variant, all the VOCs have switched the previous dominant variant into the most recent one in every geographical region. Delta variant rapidly displaced the Alpha variant in the United Kingdom and the United States by June 2021 [6]…. Consistent with our previous work, the amino acids shared among all the ACE2 binding S proteins remain constant, indicating that they are essential for the accurate binding to the receptor. The remaining amino acids are responsible for changes in the binding affinity.”

Response to comment 7

“The RMSD reflects that all the mutations in the Omicron variant, change not only the sequence but also the structure, which compromise antibodies recognition.”

Response for comment 8

“The best strategy is not to use monoclonal antibodies that bind to specific epitopes that are under selective pressures but polyclonal serum and or to elicit the host’s B cell response that could recognize different epitopes along the virus, especially those involving amino acids that are crucial for the binding to the receptor or conserved regions.”

Reviewer 2 Report

This is an excellent, carefully performed and analysed study.

Your conclusions will attract a lot of discussion. Could you please go over each sentence in your conclusion and provide evidence for each of the ideas in the following quote from your conclusions section. 

"The selected characteristics could have been the result of selection pressures induced by mass vaccination throughout the world and by the persistence or recurrent infections in immunosuppressed/immunocompromised individuals, who did not eliminate the infection and ended up facilitating the selection of viruses with characteristics different from the previous VOCs, but with the same or greater affinity. To explain how Omicron variant emerged with such an enormous amount of mutations, different hypothesis have being propounded [1]. The ancestral lineage may have been evolving i) in isolated populations ii) in immune-compromised individuals iii) in infected animals and returned to humans as a zoonotic event, or iv) by recombination of 
different lineages. Although for now the origin is not known, Omicron ancestral virus evolved independently forming a different monophyletic clade [39, 40]. Some mutations were selected in order to better adapt to the host, however, there should be an event that enhanced the mutation rate notwithstanding the most related variant, Alpha, had less mutations and a lower mutational rate [39]. The most probable situation was that Omicron ancestral virus emerged a long time ago and it has been circulating and 
evolving without being identified but under stringent selective pressures. The neutrality test of Omicron’s S protein announces the probable start of the end pandemic, indicating adaptation to the host. The more the probabilities of occurrence get closer to the ones of the S proteins of the Betacoronavirus, the more the protein approaches to the behavioral evolution of viral proteins with a long time co-evolving and adapting to 
their host." 

Author Response

May 13th, 2022

Mr. Aiden Yu
Editor vaccines

E-Mail: aiden.yu@mdpi.com

Manuscript ID: vaccines-1688687 

Title: The spike protein of SARS-CoV-2 is adapting because of selective pressures

Authors: Georgina I. López-Cortés, Miryam Palacios-Pérez, Hannya F. Veledíaz, Margarita Hernández-Aguilar, Gerardo R. López-Hernández, Gabriel S. Zamudio, Marco V. José,

Dear Dr. Aiden Yu:

Thank you for your email dated May 10th, 2022, and for the reviewer’s comments.

We start by thanking the reviewers for their careful reading of our paper and for helping us to improve its quality and presentation.

Please find enclosed with this letter the revised version of the manuscript.

We have carefully read the reviewer’s comments of our previous version of the manuscript, and we have rewritten the paper with their suggestions in mind. The reviewers raised a series of good questions. We have considered their comments modifying the paper accordingly. The modifications in the manuscript are highlighted in red color, and the answers in this reply letter are also marked in red.

We give below the specifics of our response to the reviewers and answer their criticisms in the same order as their reports.

REVIEWER 2

This is an excellent, carefully performed, and analysed study.

Your conclusions will attract a lot of discussion. Could you please go over each sentence in your conclusion and provide evidence for each of the ideas in the following quote from your conclusions section. 

"The selected characteristics could have been the result of selection pressures induced by mass vaccination throughout the world and by the persistence or recurrent infections in immunosuppressed/immunocompromised individuals, who did not eliminate the infection and ended up facilitating the selection of viruses with characteristics different from the previous VOCs, but with the same or greater affinity. To explain how Omicron variant emerged with such an enormous amount of mutations, different hypothesis have being propounded [1]. The ancestral lineage may have been evolving i) in isolated populations ii) in immune-compromised individuals iii) in infected animals and returned to humans as a zoonotic event, or iv) by recombination of different lineages. Although for now the origin is not known, Omicron ancestral virus evolved independently forming a different monophyletic clade [39, 40]. Some mutations were selected in order to better adapt to the host, however, there should be an event that enhanced the mutation rate notwithstanding the most related variant, Alpha, had less mutations and a lower mutational rate [39]. The most probable situation was that Omicron ancestral virus emerged a long time ago and it has been circulating and evolving without being identified but under stringent selective pressures. The neutrality test of Omicron’s S protein announces the probable start of the end pandemic, indicating adaptation to the host. The more the probabilities of occurrence get closer to the ones of the S proteins of the Betacoronavirus, the more the protein approaches to the behavioral evolution of viral proteins with a long time co-evolving and adapting to their host." 

We thank the reviewer for her/his comments.

We now provide the following sentences with the aim to support and clarify our conclusions:

“The mutational profile of Omicron’s spike exhibits Darwinian evolution but tends to be closer to the neutral control, more like the spike of Betacoronavirus than the previous variants Alpha, Beta, Gamma, and Delta. In contrast, the evolutionary mechanism of RBD clearly follows Darwinian adaptation.”

“Also the mutations at the RBDs’ interface changed the electrostatic properties, from neutral to positive. The receptor’s surface presents negative charged spots (Figure 5), such that both surfaces attract each other, increasing the binding affinity.”

“The best strategy is not to use monoclonal antibodies that bind to specific epitopes which are under selective pressures but polyclonal and/or to elicit the host’s B cell response that could recognize different epitopes along the virus, especially those involving amino acids that are crucial for the binding to the receptor or conserved regions.”

Reviewer 1 and 2

We have added the following relevant references

  • Harvey, W. T. et al. SARS-CoV-2 variants, spike mutations and immune escape. Nat. Rev. Microbiol. 19, 409–424 (2021).
  • Tao, K. et al. The biological and clinical significance of emerging SARS-CoV-2 variants. Nat. Rev. Genet. 22, 757–773 (2021).
  • Yin, W. et al. Structures of the Omicron spike trimer with ACE2 and an anti-Omicron antibody. Science 375, 1048–1053 (2022).
  • Tao, K. et al. The biological and clinical significance of emerging SARS- CoV-2 variants. Nat. Rev. Gen. 22, 757-773 (2021).
  • Pulliam, J. R. C. et al. Increased risk of SARS-CoV-2 reinfection associated with emergence of the Omicron variant in South Africa. Preprint at https://doi.org/ 10.1101/2021.11.11.21266068 (2021).
  • Cele, C. et al. Omicron extensively but incompletely escapes Pfizer BNT162b2 neutralization. Nature 602, 654-664. (2021). https://doi.org/10.1038/s41586-021-04387-1

All in all, we do thank the reviewers for their helpful comments and criticisms; for we feel that they have helped us to improve the quality and presentation of the paper. We expect that the present version of the manuscript answers all their concerns.

Yours Sincerely,

Marco V. José PhD

Theoretical Biology Group

Instituto de Investigaciones Biomédicas

Universidad Nacional Autónoma de México

Apartado Postal 70228 Ciudad Universitaria

C.P. 04510 Ciudad de México, México

Tel: 01-52-555- 622-3894

Mobile: +52-55-1320-6111

Email: marcojose@biomedicas.unam.mx
